

# Variation of soil microbial carbon use efficiency (CUE) and its Influence mechanism in the context of global environmental change: a review

Samuel Adingo[1], Jie-Ru Yu[2], Liu Xuelu[2], Xiaodan Li[3], Sun Jing[2] and Zhang Xiaong[1]

[1] College of Forestry, Gansu Agricultural University, Lanzhou, Gansu, China
[2] College of Resources and Environment, Gansu Agricultural University, Lanzhou, Gansu, China
[3] School of Management, Gansu Agricultural University, Lanzhou, Gansu, China

Corresponding author
Liu Xuelu, liuxl@gsau.edu.cn

## ABSTRACT

Soil microbial carbon utilization efficiency (CUE) is the efficiency with which microorganisms convert absorbed carbon (C) into their own biomass C, also referred to as microorganism growth efficiency. Soil microbial CUE is a critical physiological and ecological parameter in the ecosystem's C cycle, influencing the processes of C retention, turnover, soil mineralization, and greenhouse gas emission. Understanding the variation of soil microbial CUE and its influence mechanism in the context of global environmental change is critical for a better understanding of the ecosystem's C cycle process and its response to global changes. In this review, the definition of CUE and its measurement methods are reviewed, and the research progress of soil microbial CUE variation and influencing factors is primarily reviewed and analyzed. Soil microbial CUE is usually expressed as the ratio of microbial growth and absorption, which is divided into methods based on the microbial growth rate, microbial biomass, substrate absorption rate, and substrate concentration change, and varies from 0.2 to 0.8. Thermodynamics, ecological environmental factors, substrate nutrient quality and availability, stoichiometric balance, and microbial community composition all influence this variation. In the future, soil microbial CUE research should focus on quantitative analysis of trace metabolic components, analysis of the regulation mechanism of biological-environmental interactions, and optimization of the carbon cycle model of microorganisms' dynamic physiological response process.

## INTRODUCTION

It is the worldwide agreement to deal with climate change by jointly controlling and slowing global warming by effectively increasing carbon (C) retention and reducing C emissions in a reasonable manner (*Hoegh-Guldberg et al., 2018*). It is critical to accurately simulate and predict the interaction between global warming and the earth's ecosystems, particularly the feedback effects and mechanisms of terrestrial ecosystems on global warming, to formulate
effective measures to increase sinks (*Hicks Pries et al., 2017*). A large number of studies have found that global warming encourages the release of soil carbon, resulting in positive feedback on global warming (*Li et al., 2019*). Microorganisms, on the other hand, are increasingly being discovered to play a key role in regulating the feedback of terrestrial ecosystems to global changes, and may even alter the expected feedback effects (*Allison, Wallenstein & Bradford, 2010*; *Frey et al., 2013*). Long-term warming, for example, reduces the decomposition of soil organic carbon by inhibiting microbial biomass and enzyme activity. Microorganisms' physiological metabolic processes, as well as their responses and adaptations to changes in the external environment, have become crucial to the terrestrial ecosystem's feedback effect (*Allison, Wallenstein & Bradford, 2010*).

Soil microbes are involved in almost all material transformation processes in the soil and connect the material circulation of the soil, biosphere, atmosphere, hydrosphere, and lithosphere. The carbon utilization efficiency of soil microorganisms (Microbial carbon use efficiency, CUE), that is, the microorganisms' ability to convert absorbed carbon into biomass carbon, is directly related to their growth. The microbial CUE is set as a constant in many soil C cycle models (*Jin, Xu & Cheng, 2020*). Field observations and indoor cultivation experiments, on the other hand, contradict this hypothesis. Changes in the external environment and nutrient conditions can have a significant impact on soil microbial CUE. According to studies, soil microbial CUE increases as soil nutrient availability increases (*Manzoni et al., 2012*) and decreases as temperature rises (*Allison, Wallenstein & Bradford, 2010*). However, there is a lack of consensus on the impact of these potential factors. Water stress, for example, inhibits the growth of microorganisms and CUE, according to studies conducted on the prairies of North America (*Tiemann & Billings, 2011*). *Leizeaga et al. (2020)* discovered, however, that lowering soil water content did not affect soil microbial CUE. Furthermore, *Siebielec et al. (2020)* documented that deleterious effects of prolonged drought on plant productivity had resulted from negative impacts on microbial abundance and community structure, and the linked reduction of nutrient availability. It can also be assumed that sudden and significant changes in soil moisture, *e.g.*, intensive rain after long drought, can significantly affect the functionality of microorganisms and the processes they control (*Siebielec et al., 2020*). Previous studies indicate that alternating periods of drought and excessive soil moisture might have a strong effect on soil biology (*Young & Ritz, 2000*). According to *Gleeson et al. (2008)*, under conditions of soil saturation with water, after a long period of drought, lysis of microbial cells, connected with the release of intracellular enzymes, occurs. In such conditions, the rate of mineralization of both carbon and nitrogen increases (*Siebielec et al., 2020*). These disparate findings reflect a lack of understanding of soil microbial CUE variation and the mechanisms that influence it, limiting accurate simulation and prediction of terrestrial ecosystem feedback (*Jones et al., 2018*).

Laboratory data, conceptual and quantitative models, and, to some extent, field-based experiments are all contributing to the development of microbial CUE concepts (*Cotrufo et al., 2013*; *Abramoff et al., 2018*; *Malik et al., 2018*). However, in order to effectively incorporate agricultural C sequestration, this information must be applied to the complexities and variation of soil microbial carbon use efficiency (CUE) and its impact

mechanism in the light of global environmental change. In this light, we illustrate areas where information is missing, such as the complexities of microbial population abiotic, biotic, and interaction, which may be crucial in accurately predicting management outcomes of soil microbial CUE. We think about these uncertainties in terms of influence mechanisms that could improve CUE in soil ecosystems. Many methodological problems have recently been discussed (*Geyer et al., 2019*) but here we concentrate on the wider influences of other factors on CUE that continue to challenge C sequestration in soil ecosystems.

The purpose of this current review is to provide a comprehensive understanding of the variation characteristics of soil microbial CUE and its influencing factors, and highlights the focus of future research, by combing and analyzing the existing literature, all to improve the current earth system model and provide a theoretical foundation for scientist, researchers and relevant stake-holders to predict future climate change.

## SURVEY METHODOLOGY

To ensure an inclusive and unbiased analysis of literature and to accomplish the review's objectives, a comprehensive analysis of published articles on soil microbial carbon use efficiency was conducted using the Science Direct (http://sciencedirect.com) database, Web of Science, and Google Scholar. The following keywords were used to retrieve relevant literature: "soil microbial carbon use efficiency", "soil microorganisms", "ecological stoichiometry", "microbial community", and "nutrient limitation in soil ecosystem". While current publications between 2014 and 2019 were considered, publications that did not fall within this time period but contained critical information and were relevant to the review's objectives were also considered. Additionally, the reference lists of the retrieved literature were combed for additional pertinent publications. It is worth noting that this review presents a cross-section of studies on soil microbial carbon use efficiency and does not include all studies on the subject.

### Definition of soil microbial carbon utilization rate

Through photosynthesis, vegetation converts $CO_2$ in the atmosphere into organic matter, forming the ecosystem's net primary productivity. To realize the biogeochemical cycle of materials and energy in the ecosystem, the majority of vegetation productivity must be reduced to inorganic nutrients by decomposer-soil microbial decomposition and mineralization, and then absorbed and used by vegetation. Microorganisms' physiological metabolic process is a combination of assimilation and alienation metabolism. Microorganisms convert part of the photosynthesis of plants into microbial biomass, while the rest is released into the atmosphere *via* respiratory metabolism. Microbial carbon utilization efficiency (*Manzoni et al., 2012*; *Sinsabaugh et al., 2016*), also known as microbial growth efficiency or substrate utilization efficiency, is the efficiency with which microorganisms convert vegetation productivity into microbial biomass in this process (*Utomo et al., 2013*). Soil microbial CUE is an important ecological parameter in the soil C cycling process. It has a direct impact on the ecosystem's C retention time

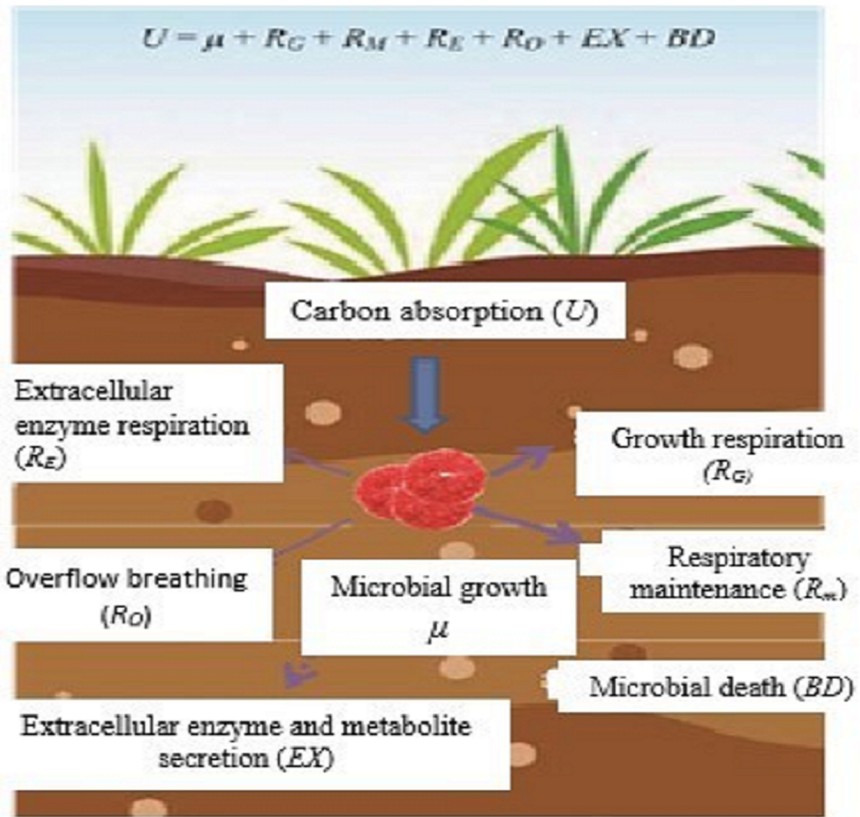

**Figure 1  Microbial metabolic components and equilibrium equation.** Sketched according to the definition of soil microorganism CUE and the mass balance equation of soil microorganism metabolism proposed by *Gleeson et al. (2008)*; U, microbial carbon absorption; μ, microbial growth; $R_G$, microbial growth respiration; $R_m$, microbial maintenance respiration; $R_E$, extracellular enzyme Respiration; $R_O$, overflow respiration; EX, secretion of extracellular enzymes and metabolites; BD, microbial death.

and turnover rate, as well as the soil's C storage capacity (*Wieder, Bonan & Allison, 2013*; *Miltner et al. 2012*; *Xu et al., 2014*).

In ecological research, microbial CUE is usually expressed as the ratio of microbial growth (μ) to absorption (U) (*Manzoni et al., 2012*; *Sinsabaugh et al., 2016*), that is, CUE = μ/U. Microorganisms absorb C from the outside world mainly for microbial growth (μ), respiratory metabolism (R), secretion of extracellular enzymes and metabolites (EX), and microbial death (BD) (Fig. 1). According to the principle of conservation of mass, microorganism U is expressed as;

$$U = \mu + R + EX + BD$$

Among them, soil microbial respiration (R) includes the respiration produced by microorganisms for growth ($R_G$), maintenance ($R_M$), extracellular enzyme production ($R_E$), and overflow process ($R_O$) (*Manzoni et al., 2012*), and is expressed as:

$$R = R_G + R_M + R_E + R_O$$
According to the definition of CUE and the mass conservation equation, the microbial CUE is expressed as:

$$CUE = \frac{\mu}{U} = \frac{\mu}{\mu + R_G + R_M + R_E + R_O + EX + BD}$$

In natural ecosystems, EX and BD are usually difficult to determine and relative to growth and respiration. The amount of EX and BD is very small and often considered negligible (*Manzoni et al., 2012*). Therefore, CUE is generally considered to be a balanced relationship between the two processes of μ and R, that is,

$$CUE = \frac{\mu}{U} = \frac{\mu}{\mu + R}$$

This definition is widely used in current microbial metabolism and soil carbon cycle models (*Manzoni et al., 2012*; *Sinsabaugh et al., 2016*).

## The primary processes governing the nature, stock, and dynamics of carbon in soils

Vertical distribution of soil organic carbon is characterized by a strong concentration gradient: from 400 g/kg in organic "O" horizons at the surface of forest soils to nearly 100 g/kg in the first cm of the organomineral horizon, with concentrations averaging less than 5 g/kg at 1 m depth. This element is found in soil at a variety of ages, ranging from a few days to several thousand years (Fig. 2) (*Balesdent et al., 2018*). Because the soil carbon stock is the sum of each previous annual input, it is contingent upon incoming carbon fluxes, biotransformation, and the duration of stabilization prior to the element's release from the soil, primarily in the form of $CO_2$ produced by decomposers' respiration.

The ratio of below-ground to above-ground biomass (root/shoot) is a highly variable indicator that is highly dependent on environmental conditions (0.1–0.3) (*Poeplau & Kätterer, 2017*). However, a significant novel finding regarding SOM mechanisms is that belowground input flux is thought to contribute more to soil organic matter than aboveground litter input *via* dead roots and rhizodeposition (*Stock et al., 2019*; *Pausch & Kuzyakov, 2018*). Rhizodeposition is the process by which living plants add carbon to the soil *via* their roots (*Pausch & Kuzyakov, 2018*; *Villarino et al., 2021*). Exudates from living roots stimulate a rapid response of soil microbes, accelerating the mineralization of native soil organic C. The amount and quality of root exudates are determined by plant species, plant age, and external factors such as biotic and abiotic stressors. Exudates from roots contain released ions (H+), inorganic acids, oxygen, and water, but are primarily composed of carbon-based compounds (*Lu, Sun & Zhu, 2017*; *Jiang et al., 2018*; *Feng et al., 2019*). Often, these organic compounds are classified into two groups: low-molecular-weight compounds such as amino acids, organic acids, sugars, phenolic acids, and a variety of secondary metabolites; and high-molecular-weight compounds such as mucilage and proteins (Table 1).

Exudates from roots have an effect on microbial communities and ecosystem functions (*Lu, Sun & Zhu, 2017*) (Fig. 3). When given a source of easily degraded carbon, such as root exudates, microbial communities accelerate the decomposition

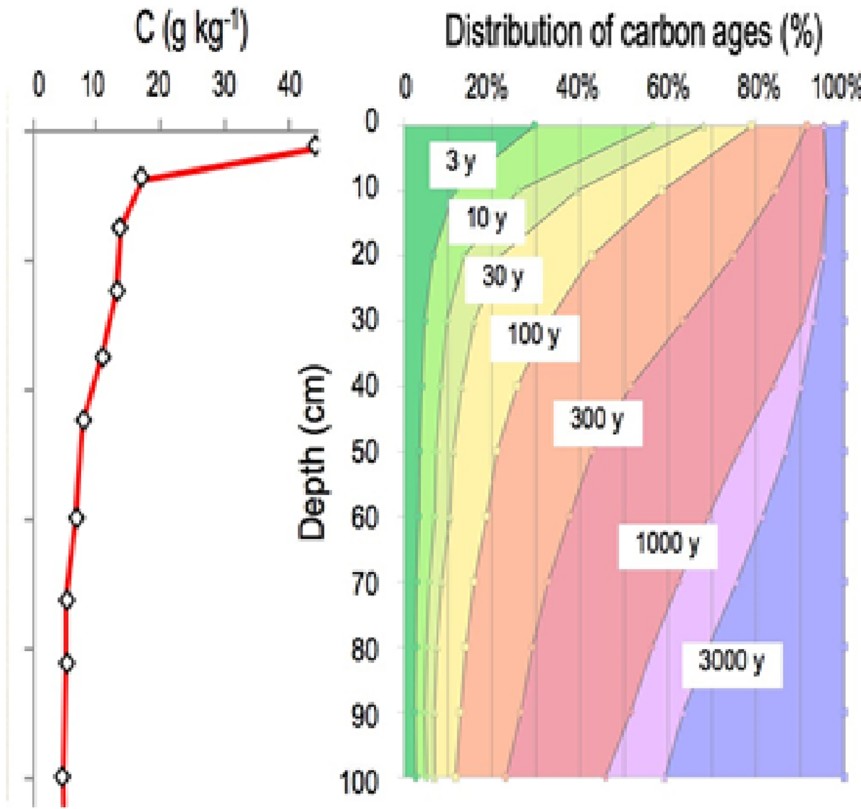

**Figure 2** The vertical distribution of organic carbon in this soil (left panel). A current distribution of carbon ages (right panel, based on data from *Balesdent et al. (2018)*.

of organic matter and mineralize nutrients that plants can use (*Moore et al., 2015*). In a laboratory study, soil bacteria and fungi increased their metabolic activity, promoting the decomposition of soil-derived and plant-derived carbon and respiration rates at low levels of simulated root exudation (*De Graaff et al., 2010*). However, at high levels of root exudation, where decomposition rates were reduced by 50%, the pattern changed. When carbon limitation is alleviated, competition for other resources among microbes may increase, and thus interactions within the microbial community may explain these counterintuitive patterns (*Moore et al., 2015*).

## Determination method of soil microbial carbon utilization

Indoor culture, in combination with a mass conservation method and a marker tracing method (*Liang et al., 2019*), is the most common method for determining soil microbial CUE (Table 2). The method of mass conservation entails directly measuring the change in mass or concentration of a substance and calculating the CUE using the principle of substance conservation (*Geyer et al., 2019*). The purpose of the marker tracking method is to effectively track the substrate's utilization path by labeling it and calculating the ratio of substrate used for growth and respiration for determination (*Scott et al., 2002*). It is currently a widely used method. Existing analysis methods can be roughly

**Table 1  Classes of compounds released in plant root exudates.**

| Class of compounds | Components |
| --- | --- |
| Carbohydrates | Arabinose, glucose, galactose, fructose, sucrose, pentose, rhamnose, raffinose, ribose, xylose and mannitol |
| Amino acids | All 20 proteinogenic amino acids, l-hydroxyproline, homoserine, mugineic acid, aminobutyric acid |
| Organic acids | Acetic acid, succinic acid, l-aspartic acid, l-glutamic acid, salicylic acid, p-hydroxybenzoic acid, p-coumaric acid. |
| Flavonols | Naringenin, kaempferol, quercitin, myricetin, naringin, rutin, genistein, strigolactone and their substitutes with sugars |
| Lignins | Catechol, benzoic acid, nicotinic acid, phloroglucinol, cinnamic acid, gallic acid, ferulic acid, syringic acid. |
| Coumarins | Umbelliferone |
| Aurones | Benzyl aurones synapates, sinapoyl choline |
| Glucosinolates | Cyclobrassinone, desuphoguconapin, desulphoprogoitrin, desulphonapoleiferin, desulphoglucoalyssin |
| Anthocyanins | Cyanidin, delphinidin, pelargonidin and their substitutes with sugar molecules |
| Indole compounds | Indole-3-acetic acid, brassitin, sinalexin, brassilexin, methyl indole carboxylate, camalexin glucoside |
| Fatty acids | Linoleic acid, oleic acid, palmitic acid, stearic acid |
| Sterols | Campestrol, sitosterol, stigmasterol |
| Allomones | Jugulone, sorgoleone, 5,7,4′-trihydroxy-3′, 5′-dimethoxyflavone, DIMBOA, DIBOA |
| Proteins and enzymes | PR proteins, lectins, proteases, acid phosphatases, peroxidases, hydrolases, lipase |

divided into methods based on microbial growth rate measurement, methods based on microbial biomass measurement, and methods based on substrate absorption rate measurement (*Mauerhofer et al., 2018*). Others include methods based on substrate concentration change determination (*Schnecker et al., 2019*) and based on different research methods and research objects (microorganisms or substrates) (*Sinsabaugh et al., 2016*). Each method has its own set of benefits, drawbacks, and application range (Table 2).

## DRIVERS OF CUE

Chemoorganoheterotrophs must make trade-offs between biomass production and energy production for each molecule of organic carbon they consume (*Müller et al., 2021*). Carbon acquisition from the environment may necessitate the production of extracellular enzymes and membrane transporters, the former of which requires energy (ATP) to construct and the latter of which may require ATP to operate. Growth also necessitates energy investment and power reduction. For example, the generation and polymerization of biosynthetic precursors into cell components require energy in the form of ATP (*Lee & Jung, 2011*), which is why the degradation of carbon storage compounds such as polyhydroxyalkanoates provides energy to the cell. Additionally, microbes must use the reducing power stored in NADH to convert a variety of food sources to biomass (*Roller & Schmidt, 2015*). The

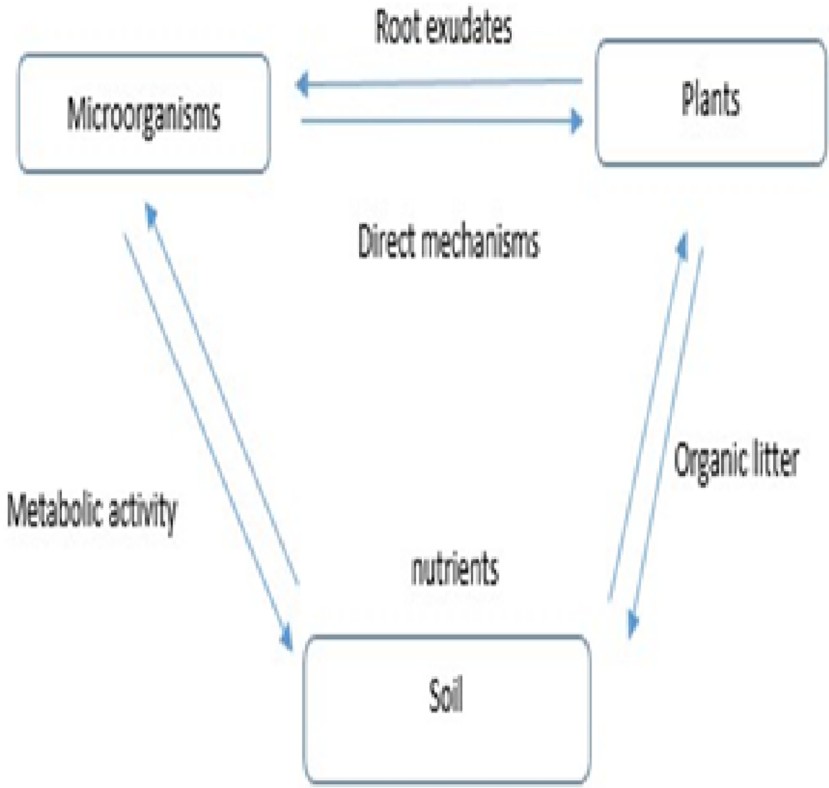

**Figure 3   Role of root exudates in ecosystem function.**

reducing power and energy required to sustain substrate uptake and metabolism are derived from the oxidation of other substrates, such that carbon taken up by a cell can be incorporated into biomass only after all of its other basic bio-energetic maintenance needs are met (*i.e.,* cell growth and thus positive CUE can occur only when there is excess carbon available above what must be used to maintain). As such, it is expected that an organism's CUE will be influenced by a variety of intrinsic and extrinsic determinants of microbial maintenance costs and cell construction requirements. Despite numerous studies examining these determinants, inconsistent methodologies have historically made it impossible to determine which intrinsic and extrinsic factors best predict CUE (*Manzoni et al., 2012*; *Sinsabaugh et al., 2013*; *Geyer et al., 2016*).

## Extrinsic influences

Carbon quality, nitrogen availability, oxygen, temperature, competition, and pH are examples of extrinsic determinants of CUE, or those that function in a way that is largely independent of organism identity. The impact of these variables on CUE can be investigated at the community level and related to factors such as maintenance costs, resource supply and demand imbalances, and discrepancies in the theoretical energy yield of various substrates (*Manzoni et al., 2012*).

Adingo et al. (2021), *PeerJ*, DOI 10.7717/peerj.12131

**Table 2  Different microbial carbon use efficiency measurement methods.**

| Measurement | Microbe Substrate | | Substrate | | Model |
|---|---|---|---|---|---|
| Measurement principle | Based on growth rate | | Based on changes in biomass | Based on absorption rate | Based on absorption rate | Based on stoichiometric ratio |
| Expression | $CUE = \frac{\mu}{\mu+R}$ | | $CUE = \frac{\Delta C_B}{\Delta C_B + R_{CUM}}$ | $CUE = \frac{U}{U-R}$ | $CUE = \frac{\Delta C_S - R_{CUM}}{\Delta C_S}$ | $CUE = \frac{\Delta A_E}{TER_{CE}}$ |
| Measurement parameters | Microbial growth rate and respiration | | Changes and accumulation of microbial biomass respiration | Substrate absorption rate and respiration | Changes in substrate concentration and cumulative respiration | Element E absorption rate, microbial C:E and the threshold element for optimal growth of microbes |
| Substrate | $^3$H-thymidine, $^3$H-leucine | $^{18}$O-H$_2$O | $^{18}$C-glucose, $^{14}$C-acetate | $^{13}$C-glucose, $^{14}$C-acetate, $^3$H-thymidine | sugars, amino sugars, amino acids and organic acids | |
| Label needed | needed | needed | needed | needed | not needed | |
| time scale | short time | short time | short time | short time | Long time | |
| Advantages | Direct measurement of microbial biosynthesis rate | Direct measurement of microbial biosynthesis rate | Simple and easy to operate | Consider microbial productivity flow | Consider the loss of microbial productivity | No measurement required |
| Disadvantages | Unsuitable for soil | Only suitable for short-term determination | Need to be converted into biomass, overestimating CUE | Only suitable for short-term determination | Need to measure the adsorption of the substrate and provide a high concentration of the substrate | There are model assumptions, empirical coefficients |
| Application field | Waters | land | land | land | land | |
| Reference | *Hoegh-Guldberg et al. (2018)*, *Hoegh-Guldberg et al. (2018)*, *Li et al. (2019)* | *Allison, Wallenstein & Bradford (2010)*, *Frey et al. (2013)* | *Jin, Xu & Cheng (2020)*, *Manzoni et al. (2012)* | *Tiemann & Billings (2011)* | *Leizeaga et al. (2020)*, *Siebielec et al. (2020)* | *Young & Ritz (2000)* |

**Notes.**

CUE, carbon utilization; $\mu$, microbial growth rate; R, total microbial respiration rate; U, substrate absorption rate; $\Delta C_B$, change in microbial biomass; $\Delta C_S$, change in substrate concentration; $R_{CUM}$, cumulative respiration rate; $A_E$, The absorption efficiency of element (E); BC: E, the ratio of C: E of microbial biomass to the C; $TER_{C:E}$, C: required for optimal growth of microorganisms.

## Quality of the substrate

As a source of food, microorganisms absorb soil organic carbon and vegetation debris. The quality of the substrate will have a considerable impact on the microbial CUE in the soil (*Jones et al., 2018*). This effect is influenced by the substrate's diverse material compositions, the decomposition process, the degree of reducibility, and the effectiveness. The ability of a substrate to be incorporated into biomass is a crucial factor of CUE. Polymeric substrates like lignin and cellulose must be depolymerized before they can be taken up by the cell, which implies a cell's resources must be reallocated from growth to enzyme production (*Allison, 2014*). The resulting monomers and dimers must be taken up by the cell after this depolymerization stage. While some molecules, such as glycerol and ethanol, can readily diffuse across the membrane, others must be transported into the cell by transport proteins, either passively or *via* proton exchange (*Whalley, Walters & Hammond, 2018*). As a result, some carbon sources demand more resources to acquire than others. Compounds enter metabolism at different stages after entering the cell, and are thus allocated to biomass (anabolism/assimilation) or energy production in different ways (catabolism). As a result, as a proxy for carbon quality, community level CUE in soil demonstrated a positive association with the cellulase:phenol oxidase enzyme potential (*Takriti et al., 2018*). If highly oxidized chemicals like oxalic acid are to be integrated into biomass (*Hervé et al., 2016*), they must expend a lot of reducing power (NADH), and they also produce very little energy compared to glucose. As a result, on oxalic acid or phenolic substances, the CUE of soil microbial communities is significantly smaller than on glucose (*Frey et al., 2013*). Other characteristics, such as whether they block other metabolic pathways (*Gazizova, Rakhmatullina & Minibayeva, 2020*), where compounds reach central metabolism, whose cell components they are transformed to *Gunina et al. (2014)*, and how quickly those components are recycled, are likely to be significant for CUE (*Dudley, Karim & Jewett, 2015*). Finally, the effect of carbon quality on CUE may be influenced by the bacteria's past nutrient regime; bacteria that have been exposed to carbon-limited settings can metabolize a far broader range of substrates than those that have been exposed to carbon-rich settings (*von Stockar & van der Wielen, 2013*). When a non-assimilable energy source is combined with a low-energy carbon source, co-metabolism can help to boost carbon conservation (*Dudley, Karim & Jewett, 2015*). Nonetheless, CUE is expected to rise when a compound's degree of reduction decreases.

The degree of C reduction of the substrate ($\gamma S$) is another important factor that affects the CUE of soil microorganisms. The degree of C reducibility refers to the chemical energy per mole of C and is usually expressed by the electron equivalent per mole of C. The $\gamma S$ of the main substrate used by microorganisms is usually in the range of 3-5 (such as organic acids, glucose, carbohydrates, and lipids), which is equivalent to the degree of C reduction of soil microorganisms ($\gamma B \approx 4.2$) (*Roels, 1980*). When the $\gamma s$ of the substrate is lower than the microbial $\gamma B$, the CUE of soil microorganisms is lower because the energy per unit of the substrate cannot meet the energy requirement of a unit of biomass production (*Roels, 1980*; *P. Gommers et al., 1988*). The analysis results of 16 substrates CUE with different degrees of reduction showed that sugar CUE (0.667) > amino sugar CUE (0.601) > amino acid CUE (0.551) > organic acid CUE (0.498).

## Temperature

CUE is frequently reported to be reduced as the temperature is raised (*Frey et al., 2013*; *Manzoni et al., 2012*). The exact process is unknown, although one theory is that microbial maintenance costs rise with temperature due to higher protein turnover (*Manzoni et al., 2012*), the necessity for lipid saturation in the cell membrane (*Hall et al., 2010*), or a heat stress response (*Manzoni et al., 2012*). Another reason could be because the amount of energy-conserving sites in the electron transport chain is temperature-dependent. Alternatively, greater temperatures may favor the desorption of chemically labile, high C:N molecules from mineral surfaces, resulting in a rise in CUE in soil (*Hilasvuori et al., 2013*). According to other researchers, temperature has no effect on intrinsic CUE (*Dijkstra et al., 2011a*), and the apparent drop in CUE with warming could be attributable to methodological errors or higher microbial turnover (*Hagerty et al., 2014*). Microbes can adjust to local temperature, according to more recent community-level research. Because growth outpaces respiration in this process, CUE should rise in tandem with temperature (*Ye et al., 2020*). Reduced substrate supply, even with moderate temperature increases, may be at the root of this, effectively starving bacteria (*Hagerty et al., 2014*). As a result, the effect of CUE on higher temperatures appears to be dependent on microbial physiology, substrate selection, and measurement method. (*Berggren et al., 2010*) have found that soil microbial CUE has a negative feedback on temperature increase, with CUE decreasing as temperature rises. This is because, when the temperature is controlled, microorganisms' growth and metabolic rate increase as the temperature rises (*Wetterstedt & Agren, 2011*). However, the temperature sensitivity of microbial respiration metabolism is higher than that of growth response (*Frey et al., 2013*), and microbial respiration increases faster than microbial growth, reducing CUE (*Veach & Griffiths, 2018*). According to *Steinweg et al. (2008)*, soil microbial CUE decreased by about 0.009 for every 1 °C increase in temperature. Under high-temperature stress, the negative feedback effect of microbial CUE is more pronounced (*Sinsabaugh et al., 2013*). According to the simulation results, 30 years of continuous temperature rise has reduced the proportion of absorbed C used for microbial growth, lowering the CUE from 0.31 to 0.23 (*Allison, Wallenstein & Bradford, 2010*). However, some studies have found that the soil microbial CUE does not change significantly as the temperature rises (*Hagerty et al., 2014*; *Dijkstra et al., 2011*). The composition of the substrate and the metabolic stage influence the response of soil microbial CUE to temperature. The CUE of soil microorganisms under the supply of a single-molecule structure substrate decreases with increasing temperature, while the CUE of soil microorganisms under the supply of a polymer structure substrate does not change with increasing temperature, according to (*Öquist et al., 2017*). Long-term warming, according to some studies, will make microorganisms adaptable. Long-term warming will cause microorganisms to reduce their basal respiration rate (*Tucker et al., 2013*). The continuous warming experiment in Harvard Forest revealed that a 5 °C increase in temperature over 18 years reduced the degree of soil microbial CUE, with an increasing temperature lower than the warming effect of two consecutive years (*Frey et al., 2013*). Because microorganisms' thermal adaptability is linked to changes in microbial community composition, reduced nutrient availability, and changes in microbial metabolic pathways, as well as substrates and

observation methods, there are still many unknowns about how microorganisms respond to temperature.

## Nitrogen availability (substrate C:N ratio)

The C:N ratio of the substrate appears to be an unambiguous driver of CUE for both isolates and soil communities. Biomolecules have certain carbon and nitrogen content ranges, and cells require a particular percentage of these biomolecules to function. Because elements in the substrate consumed by a microbe are rarely available in the exact ratios required to maintain and create new biomass, nitrogen or carbon will be mineralized first to restore the proper elemental ratio (*Mooshammer et al., 2014*). As a result, low CUE is likely to be related with substrates with high C:N ratios or nitrogen limiting circumstances (*Manzoni et al., 2012*).

Microorganisms will respond to nutrient changes by changing the carbon assimilation pathways regulated by their chemical enzymes when the availability and composition of nutrients changes (*Traoré et al., 2016*). Some studies have found that as nutrient availability increases, soil microbial CUE increases (*Manzoni et al., 2012*). The microbial carbon absorption rate and nutrient concentration have a saturation function relationship. Microorganisms will maintain the optimal carbon absorption rate to meet the absorption system's resource consumption when resources are limited (*Sinsabaugh & Shah, 2012*; *Hobbie & Hobbie, 2012*). When the availability of nutrients increases and the nutrient concentration exceeds the microorganisms' equilibrium concentration, carbon absorption increases, increasing CUE. Nutrient restriction, on the other hand, will lower CUE. The microorganism's decomposition and synthesis, as well as the coupled metabolic process, will be altered by nutrient restriction, which will increase metabolites such as extracellular enzymes and polysaccharides, and a decrease in CUE (*Kuang et al., 2016*; *Wang, Bhardwaj & Webster, 2017*). This has also been proven by a large number of N addition experiments. The addition of nutrients such as nitrogen can either stimulate or inhibit microorganism activity and respiration metabolism.

Soil microorganisms will adjust and redistribute resources within cells to meet the demand for a variety of nutrient elements, which will further affect microorganism growth and CUE (*Kuang et al., 2017*; *Kaiser et al., 2014*). When microorganisms are restricted by a nutrient, they will expend more energy to obtain the missing nutrient elements, inhibiting microorganism growth and CUE (*Manzoni et al., 2012*). When microorganisms are restricted by the P element, for example, they will increase the input for P element resource acquisition, lowering CUE (*Sinsabaugh et al., 2013*). Experiments have shown that nutrient deficiency inhibits microorganism growth and lowers CUE (*Öquist et al., 2017*). This nutrient scarcity is usually the result of the availability of multiple nutrients being constrained at the same time (*Sinsabaugh et al., 2013*).

## Availability of oxygen

The presence of oxygen, as the most powerful biological oxidant and the best terminal electron acceptor, can play a key role in CUE. Oxygen's high redox potential allows it to catch electrons that have traveled longer along the electron transport chain, resulting in

a higher electrochemical gradient than other terminal electron acceptors (*Delattre et al., 2019*). In the absence of oxygen, organisms must either ferment molecules (using internal electron acceptors) or employ terminal electron acceptors with a lower redox potential, such as nitrate. When organisms can't use oxygen, the quantity of energy they can acquire from a particular substrate is lowered.

Anaerobic circumstances have been found to lower the percentage of carbon traveling *via* the anabolic pentose phosphate pathway and increase the percentage going *via* glycolysis, in addition to their effect on direct oxygen-dependent metabolic pathways (*Dijkstra et al., 2011a*; *Dijkstra et al., 2011b*). Although low oxygen conditions are predicted to limit biomass yield, because $CO_2$ is the sole waste product generally assessed, lower CUE will not necessarily be observed. In addition, while the energy yield from a substrate may be higher under aerobic conditions, the cost of biosynthesis and cell maintenance (for example, due to oxidative damages) is lower under anaerobic settings (*Hoehler & Jørgensen, 2013*). As a result, predicting a general influence of oxygen availability on CUE is difficult.

## Competition and interconnectedness

Cheating strategy is an important aspect of competitiveness that is crucial to CUE. Enzyme manufacturing demands a significant expenditure of resources, thus releasing them into the environment, where they and their substrates may be hijacked by other cells, is a risky proposition. As a result, certain bacteria may "cheat" by not producing extracellular enzymes and instead rely on the monomers created by the enzymes of other organisms. When privatization of resources is not allowed, such cheating is temporarily favored, but it eventually leads to enzyme producers discontinuing to make enzyme when they are less likely to receive the benefits of their investment (for example, in a well-mixed environment) (*Allison et al., 2014*). Because decreasing soil moisture reduces enzyme transport, extracellular enzyme synthesis is projected to increase. Interactions between species, on the other hand, may result in greater CUE if cross-feeding happens. Amino acids, for example, can be categorized depending on precursor demands, with some amino acids synthesized more efficiently from gluconeogenic substrates and others produced more efficiently from glycolytic substrates (*Waschina et al., 2016*). As a result, it is possible that an organism fermenting glucose and making amino acids may transfer gluconeogenic substrates like lactate to another organism, which would then use it to generate amino acids and then share the rest with the original donor.

Because different microbial populations decompose and absorb the organic matter at different rates, the structure and composition of the microbial community have an impact on soil microbial CUE (*Freixa et al., 2016*; *Ziegler & Billings, 2011*). Fast-growing microorganisms that use the 'opportunity' growth strategy are more adapted to high-nutrient environments and have lower CUE than slow-growing microorganisms (*Keiblinger et al., 2010*; *Štursová et al., 2012*). The CUE of a microbial community dominated by fungi is often higher than that of a microbial community dominated by bacteria, according to *Yang et al. (2020)*. Fungi have a wider C:N:P variation range than bacteria, and their C: N ratio is higher than bacteria's, which has a higher requirement for C and has a high CUE (*Keiblinger et al., 2010*). However, some studies have found that the CUE of soil communities does

not differ significantly (*Utomo et al., 2013*). (*Thiet, Frey & Six, 2006*) found no significant difference in CUE between communities with a high fungi/bacteria ratio and communities with a low fungi/bacteria ratio, which was $0.59 \pm 0.02$ and $0.61 \pm 0.01$, respectively, in a study of farmland ecosystems. Increased substrate C: E can also increase the CUE of the fungal community while decreasing the CUE of the bacterial community, according to studies (*Keiblinger et al., 2010*). The microbial community's interspecific competition will reduce the microorganisms' CUE (*Maynard, Crowther & Bradford, 2017*). Because the microbial community's composition is highly susceptible to changes in the external environment and human activities, as well as changes in the substrate's quality and composition, quantifying the differences in the composition of different communities remains a work in progress (*Utomo et al., 2013*; *Keiblinger et al., 2010*).

## pH

pH is hypothesized to influence CUE since it is both a significant determinant of bacterial community structure and a possible stressor. *Sinsabaugh et al., (2016)* discovered a weak but significant CUE minimum at a pH of 5.4 in a meta-analysis of global soils, which they attributed to changes in the bacterial to fungal ratio. pH is also thought to alter the availability of nutrients and harmful metals like aluminum, which could impair CUE indirectly by redirecting resources to stress response rather than growth. At high pH, the necessity to produce novel antiporters or modify metabolism to accommodate additional organic acids may impart direct impacts on CUE (*Barberán et al., 2017*). However, pH optima are not only found in bacteria isolated from the same soil, but they are also phylogenetically preserved (*Liu et al., 2016*). This suggests that the degree of stress reaction (and, as a result, reduction in CUE) that bacteria exhibit in response to pH is likely to vary between organisms while being phylogenetically conserved. In general, soil habitats with pH values in the neutral range have more bacterial diversity than those with pH values that are more acidic or alkaline (*Wang, Liu & Bai, 2018*). Fungal populations, on the other hand, may be less sensitive to pH changes (*Wang et al., 2020*). pH changes can affect the solubility of many soil constituents, such as metals (*Basta, 2004*). The increase in free metals in wastewater-irrigated soil was linked to a drop in soil pH which, in turn, can influence microbial populations (*Lucchini et al., 2014*).

## CUE "intrinsic" determinants and indicators

The traits encoded in a cell's genetic or epigenetic imprint are known as "intrinsic" determinants of CUE. Although many of these intrinsic determinants are predicted to be influenced by the extrinsic factors mentioned above (*Waschina et al., 2016*), identifying genetic drivers of CUE should help interpret carbon cycling data in the context of environmental metaomic data. Many of these parameters have been investigated in the context of growth yield in model organisms like E. coli, which is related but not identical to CUE.

## Phylogenetic determinants of carbon usage efficiency

Some scholars have claimed that disparities in CUE exist at high taxonomic levels. Fungi-dominated communities, for example, have been shown to be more efficient than

bacteria-dominated communities (*Sinsabaugh et al., 2016*), likely because their biomass has a greater CN ratio. Fungus-dominated communities, on the other hand, may simply be able to access alternative carbon stores than bacteria (*Soares & Rousk, 2019*) and/or be found in soils with edaphic characteristics that favor higher CUE (*Sinsabaugh et al., 2016; Soares & Rousk, 2019*). Alternatively, CUE in bacteria may react to different abiotic stimuli than in fungi (*Keiblinger et al., 2010*). If different groups of organisms have different maximum attainable CUEs, it could be due to genetically encoded physiological constraints, such as the need for carbon allocation to abundant peptidoglycan in Gram positive organisms, transporters spanning both membranes in Gram negative bacteria, or abundant intracellular membranes (*Lee et al., 2009*). Despite these significant differences in cell chemistry and biosynthetic precursor needs, metabolic modeling suggests that CUE on glucose is unaffected by the Gram positive: Gram negative: fungal ratio (*Dijkstra et al., 2011a*). As a result, phylogeny may have an impact on CUE's ability to respond coherently to environmental variables such as temperature, drought, or nitrogen availability (*Amend et al., 2016; Morrissey et al., 2016*).

The number of copies of the ribosomal RNA operon may also be a significant predictor of CUE (rrN). In bacteria, rrN establishes the highest limit on growth rate (*Klappenbach, Dunbar & Schmidt, 2000*), allowing up to 75% of transcriptional effort to be dedicated to producing ribosomes during rapid growth. rrN is phylogenetically conserved to the point where its presence in unsequenced genomes may be anticipated based on the values of close relatives (*Kembel et al., 2012*). Acidobacteria and other "oligotrophic" phyla can grow slowly in nutrient-poor environments and have low rrN, whereas fast-growing copiotrophic taxa like Betaproteobacteria and Bacteroidetes have high rrN (*Lupwayi et al., 2017*). It's unclear whether rrN is a CUE determinant in and of itself, or merely a stand-in for other conserved CUE factors.

## Interplay between biotic and abiotic drivers of CUE

The multiple biotic and abiotic elements mentioned above may influence CUE both alone and in combination with other factors (Fig. 4). Taxa growing on the same mixed substrate media, for example, may have different CUEs due to their preference for organic acids over sugars (*Deutscher, Francke & Postma, 2006*). CUE may be influenced by interspecific competition as well as nitrogen supply (*Maynard, Crowther & Bradford, 2017*).

The majority of studies on the impact of climate/environmental change on biological systems and soil microbes to date have focused on specific elements like increased atmospheric $CO_2$ concentrations, warmth, or drought. Interactions between these components, on the other hand, have the potential to have additive or antagonistic impacts on soil microorganisms and their activities connected to greenhouse gas production (*Stefan et al., 2021*). The impacts of multiple and interacting climate drivers on soil microbes and their contribution to climate change are poorly understood, and because they are so complicated, they are likely to be difficult, if not impossible, to predict (*Ochoa-Hueso et al., 2019*). While some studies show unpredictably responses of soil microbial communities and their activities to the combined effects of elevated temperature and atmospheric $CO_2$ (*Deltedesco et al., 2020*), others show strong additive effects with significant potential
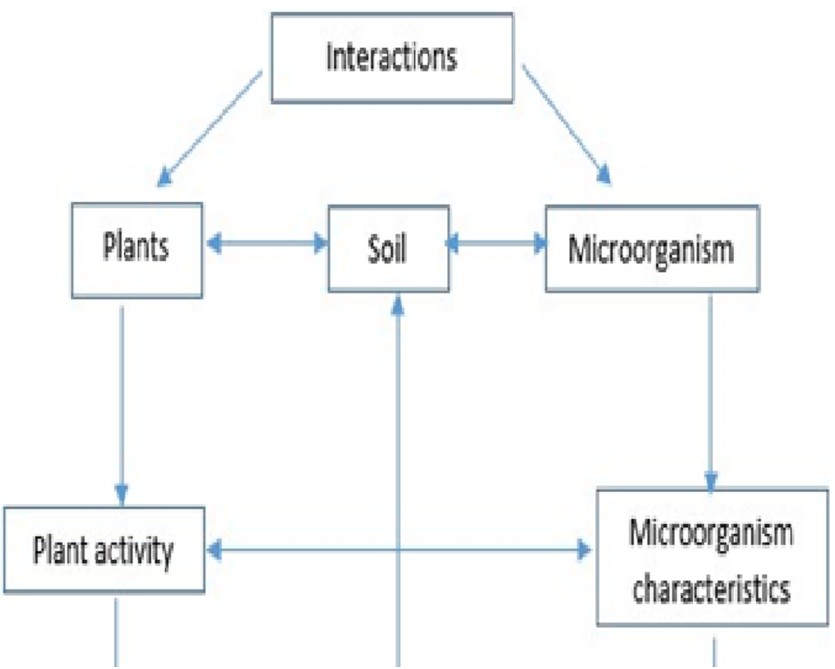

**Figure 4    Linkage between environmental factors, plants and soil and regulations of soil microbial processes.**

for carbon exchange feedback. For example, the combined and positive effect of elevated temperature and atmospheric $CO_2$ on microbial decomposition of peat was found to be greater than when these factors operated separately (*Kuzyakov, Bogomolova & Glaser, 2014*), resulting in an even stronger positive feedback on carbon loss from soil as DOC and respiration.

Other global change phenomena such as N deposition, invasion of new species, and land use change, all of which have the ability to alter soil microbes through a number of direct and indirect pathways, but also interact with climate change, make the picture even more complicated. N enrichment, for example, has direct and differential effects on extracellular enzymes involved in decomposition processes (*Zhang et al., 2018*), as well as the abundance and diversity of bacteria, saprophytic fungi, and mycorrhizal fungi (*Zhang et al., 2018*). N deposition can also have an indirect impact on soil microbes and decomposition processes by changing vegetation composition and productivity (*Zhang et al., 2018*) and alleviating progressive N limitation of plant growth, which is common when atmospheric $CO_2$ levels are high. Although little is known about the impact of combined global changes on soil microbial populations, they certainly have the ability to enhance, inhibit, or even cancel climate change-related effects on soil microbes and their carbon their CUE. To understand soil microbial reactions to global changes and their ramifications for carbon cycle feedbacks, future studies should employ a multifactor experimental strategy.

## Variability of soil microbial carbon utilization rate

Soil microbial CUE is not constant in natural ecosystems. Microorganisms would only assimilate organic matter and completely assimilate the substrate in an ideal state, so the carbon utilization rate would be 1. Microorganisms' growth, on the other hand, is always accompanied by the consumption of respiratory metabolic products. Microorganisms have a maximum CUE ($CUE_{max}$) of less than 1 due to thermodynamic constraints (*Roels, 1980*). Microorganisms' actual growth is governed by their stoichiometric balance, which shows that CUE varies with changes in essential element absorption efficiency (E), the microorganisms' C: E ratio, and the optimal growth C: E ratio threshold. When the nitrogen absorption efficiency $A_N$ =1, the soil microbial CUE can approach the maximum CUE when the C: N ratio threshold $TER_{C:N}$ is15 ($CUE_{max} = 0.6$); when the $TER_{C:N}$ is 30, the soil microbial CUE drops to 1/2 of $CUE_{max}$ ($CUE_{max}/20.3$). Because $A_N$ is usually less than one in the real world, it's difficult for microorganisms' actual CUE to reach $CUE_{max}$. The results of three different methods of ATP generation, electron transfer, and energy conversion show that the actual maximum CUE of microorganisms is around 0.6 due to thermodynamic limitations (*Roels, 1980*).

The results of the experiments show that the microbial CUE in different soil layers varies. For example, the CUE of microorganisms in the mineral layer (0.2840.005) is higher than the CUE of microorganisms in the organic layer (0.2050.008) (*Sinsabaugh et al., 2017*). The CUEs of different microbial populations are also different. Based on an integrated analysis of experimental observation data, (*Sinsabaugh et al., 2015*) discovered that the CUE of bacteria is around 0.3360.213, which is higher than that of fungi (0.3260.196). The CUE of soil microbial varies depending on the type of vegetation. The research and study of (*Takriti et al., 2018*) on Siberian vegetation transects revealed that the CUE of soil microorganisms decreased as they progressed from the meadow steppe to the Taiga forest and tundra. Soil microbial CUE is usually set as a parameter in the current large number of biogeochemical cycle models, with values ranging from 0.25 to 0.6 (Table 3). Because different substrate compositions and other influences are taken into account in different biogeochemical models, the CUE parameter values vary. For example, the decomposition of underground and above-ground organic matter in the CENTURY model uses different soil microbe CUs of 0.45 and 0.55, respectively (*Parton et al., 1987*). The Daisy, NCSOL, ICBM, and other models take carbon pool activity into account. CUE is 0.6 for activated carbon pools and less than 0.6 for inert carbon pools (*Hansen et al., 1991*; *Kätterer & Andrén, 2001*; *Molina et al., 1983*) (Table 3). However, *Manzoni et al. (2018)* pointed out that measured soil microbial CUE results are frequently lower than the model's preset value, implying that the current model understates the true hetero-oxygen respiration flux to some extent.

## C-use efficiency across spatial and temporal scales

Integrating C exchange rates in space and time is required to move up geographical and temporal scales. Integrating these exchange rates, in turn, essentially averages out the contributions at smaller or shorter scales by taking into account a larger number of organisms (*e.g.*, populations *vs.* individuals), a broader geographic region, and longer time periods. When compared to smaller sizes, this averaging effect often results in a lower CUE.

**Table 3 Variation of microbial carbon use efficiency.**

| Type of model | Carbon use efficiency | Attributes | References |
|---|---|---|---|
| Measured values | 0.39 | | *Hoegh-Guldberg et al. (2018)* |
| | 0.58 | | *Hicks Pries et al. (2017)* |
| | 0.44–0.73 | | *Li et al. (2019)* |
| | 0.58–0.70 | | *Allison, Wallenstein & Bradford (2010)* |
| | 0.26–0.68 | | *Frey et al. (2013)* |
| | 0.45–0.75 | | *Jin, Xu & Cheng (2020)* |
| | 0.46–0.62 | | *Manzoni et al. (2012)* |
| | 0.35–0.83 | | *Tiemann & Billings (2011)* |
| | $0.24 \pm 0.08$ | | *Leizeaga et al. (2020)* |
| | 0.49–0.79 | | *Siebielec et al. (2020)* |
| | 0.42–0.84 | | *Young & Ritz (2000)* |
| Stoichiometric model | 0.29 | | *Gleeson et al. (2008)* |
| Q-model | 0.25 | | *Jones et al. (2018)* |
| CENTURY-model | 0.45 | Decomposition of underground organic matter | *Cotrufo et al. (2013)* |
| | 0.55 | Decomposition of surface organic matter | *Cotrufo et al. (2013)* |
| Daisy, NCSOIL, ICBM model | 0.6 | Activated carbon pool | *Abramoff et al. (2018)*, *Malik et al. (2018)* and *Geyer et al. (2019)* |
| Daisy, NCSOIL, ICBM model | <0.6 | Most inert carbon pools | *Abramoff et al. (2018)*, *Malik et al. (2018)* and *Geyer et al. (2019)* |

CUE is calculated on a variety of geographic and temporal dimensions depending on the system of interest, necessitating interpretation of CUE in terms of averaged C exchange rates at different scales. Because organism-level CUE estimations are skewed toward actively growing individuals who are often separated in highly controlled environments, spatial averaging under field settings, which includes dormant or slowly growing individuals, resulting in lower population- or community-level CUE. Individual plants have a CUE of roughly 0.6, but plant communities have a GGE of roughly 0.4.

When comparing CUE of microbial isolates and soil microbial communities, which are not statistically distinct, this disparity between CUE estimates at individual and community scales is not visible. Despite the presence of high values in some communities, the CUE of aquatic microbial communities is much lower than that of microbial isolates (CUE0.25). The high CUE of soil microbial communities could be related to soils having more resources than aquatic habitats, or to soil amendment using labile chemicals that encourage microbial activity while masking the contribution of slow-growing organisms (*Sinsabaugh et al., 2013*). Short-term measurements, such as those taken after adding labile substrates to heterotrophic systems or during busy growing phases for plants, tend to lower CUE. Long-term CUE, on the other hand (if biomass turnover is properly accounted for), includes periods of poor growth caused by inappropriate environmental conditions, during which maintenance costs remain high as growth stagnates. This could explain why

litter microorganisms' long-term CUE is lower than microbial CUE assessed over short durations in other systems.

## Limits of ecological environmental factors to Soil microbial CUE

Temperature, humidity, precipitation, and soil moisture all influence microbial metabolism, altering the balance between and R and thus affecting CUE (*Manzoni et al., 2012*) (Fig. 4). *Xu et al. (2014)* have found that soil microbial CUE has a negative feedback on temperature increase, with CUE decreasing as temperature rises. This is because, when the temperature is controlled, microorganisms' growth and metabolic rate increase as the temperature rises (*Utomo et al., 2013*). However, the temperature sensitivity of microbial respiration metabolism is higher than that of growth response (*Manzoni et al., 2012*), and microbial respiration increases faster than microbial growth, reducing CUE (*Allison, Wallenstein & Bradford, 2010*). According to *Steinweg et al. (2008)*, soil microbial CUE decreased by about 0.009 for every 1 °C increase in temperature. Under high-temperature stress, the negative feedback effect of microbial CUE is more pronounced (*Berggren et al., 2010*). According to the simulation results, 30 years of continuous temperature rise has reduced the proportion of absorbed C used for microbial growth, lowering the CUE from 0.31 to 0.23 (*Allison, Wallenstein & Bradford, 2010*). However, some studies have found that the soil microbial CUE does not change significantly as the temperature rises (*Hagerty et al., 2014*). The composition of the substrate and the metabolic stage influence the response of soil microbial CUE to temperature. The CUE of soil microorganisms under the supply of a single-molecule structure substrate decreases with increasing temperature, while the CUE of soil microorganisms under the supply of a polymer structure substrate does not change with increasing temperature, according to *Öquist et al. (2017)*. Long-term warming, according to some studies, will make microorganisms adaptable. Long-term warming will cause microorganisms to reduce their basal respiration rate (*Tucker et al., 2013*). The continuous warming experiment in Harvard Forest revealed that a 5 °C increase in temperature over 18 years reduced the degree of soil microbial CUE, with an increasing temperature lower than the warming effect of two consecutive years (*Frey et al., 2013*). Because microorganisms' thermal adaptability is linked to changes in microbial community composition, reduced nutrient availability, and changes in microbial metabolic pathways, as well as substrates and observation methods, there are still many unknowns about how microorganisms respond to temperature and how they do so.

Another important environmental factor that influences microorganism growth and respiration, and thus CUE, is soil moisture and water availability (*Tiemann & Billings, 2011*) (Fig. 4). The effect of soil water availability on CUE is complex and variable, similar to the effect of temperature, and is influenced by intensity, duration, soil type, and soil water stress. Short-term water stress stimulates a microbial response to water stress, promotes microbes to reduce the impact of drought by increasing osmotic pressure regulation or short-term dormancy, and increases soil microbial CUE, according to studies (*Tiemann & Billings, 2011*). Long-term water stress, on the other hand, reduces the solubility and absorption of soil substrates, inhibiting microorganism growth (*Manzoni et al., 2012*). Long-term water stress, on the other hand, will increase the metabolic consumption

of microorganisms, lowering CUE (*Tian et al., 2019*). The CUE of soil microorganisms in an anaerobic environment is lower than in an aerobic environment, according to studies (*Burgin et al., 2011*). The metabolites of soil microorganisms are released from $CO_2$ to CH4, which cannot be completely oxidized, and CUE decreases in an anaerobic environment (*Burgin et al., 2011*).

## Impact of wastewater irrigation on microbial community and processes in the soil

Soil microbial communities are formed by a complex web of interrelationships between abiotic (physical and chemical soil qualities) and biotic (microbial community composition) elements (macro- and microbiological soil components). The effect of wastewater on soil microbial communities is thought to be dependent on direct external microbiota inputs, which, in the unlikely worst-case scenario, would result in the extinction of autochthonous bacteria due to competition. Furthermore, and no less important, are the indirect effects of wastewater, which may lead to changes in physicochemical soil qualities and, as a result, microbial community disturbances. Both types of effects are largely unknown, resulting in significant knowledge gaps. The following sections cover the direct and indirect effects of wastewater irrigation on soil microorganisms, processes, and soil characteristics.

## Salinity

In comparison to freshwater, wastewater has larger amounts of dissolved inorganic compounds, such as soluble salts. As a result, wastewater irrigation may enhance soil salinization (an increase in the concentration of soluble salts) or sodification (increase of sodium ions relative to other cations). Salinization is linked to an increase in electrical conductivity. Sodification, on the other hand, has a detrimental impact on the stability of soil aggregates and soil structure, resulting in increased soil compaction, decreased soil permeability, and decreased hydraulic conductivity (Table 4). The following are the most often cited negative consequences of wastewater irrigation (Table 5). The principal effects on microbial communities are related to changes in soil structure and a decrease in osmotic potential (*Bandopadhyay et al., 2018*). Increased soil salinity has been demonstrated to diminish fungal and bacterial counts, as well as microbial diversity and biomass (*Ke et al., 2013*). Salinity and sodicity were linked by *Trelka et al. (2016)* to increased microbial stress and a decrease in the metabolic efficiency of the microbial population. Indeed, soil salinity appears to influence C and N mineralization as well as nitrification retardation.

## Texture of the soil

While wastewater irrigation increased aggregate stability in loamy sand soils, it decreased aggregate stability in sandy clay and clay soils (*Levy et al., 2014*). Because 40–70% of soil bacteria are associated with stable micro aggregates and clay particles smaller than 20 m, the properties of the soil aggregates are crucial (*Lin & Gan, 2011*). As a result, wastewater irrigation's involvement with the production of soil aggregates is likely to change soil microhabitats and, as a result, influence soil microbial populations (*Trivedi et al., 2019*). Increases in soil organic matter were found to have a positive impact on

**Table 4 Examples of soil physicochemical, biochemical and microbiological properties influenced by the variation of selected parameters.**

| Parameter | Effect in the soil/environment | Effect on microbiological parameter | Reference |
|---|---|---|---|
| pH | Availability of nutrients and trace elements | Community richness and diversity | |
| | Mineralization of organic matter | | *Hoegh-Guldberg et al. (2018)* |
| | Cation exchange capacity | | |
| Organic matter | Aggregate formation and stabilization of soil structure | Selection of specific population | |
| | Water retention | Soil microhabitats | *Hicks Pries et al. (2017)* |
| | Enzymatic activity | | |
| | Availability of organic and inorganic contaminants | | |
| Nutrients (N, P, K) | Improvement of soil fertility | Disturbance of soil microbial communities | |
| | Increase of soil organic matter | Microbial catabolic activity | *Li et al. (2019)* and *Allison, Wallenstein & Bradford (2010)* |
| | Water retention | | |
| | Leaching to groundwater and risk of eutrophication of aquatic systems | | |
| Salinity | Soil salination or sodification | Soil microhabitats | |
| | Decrease soil aggregate and structure | Community diversity and activity | *Frey et al. (2013)* and *Jin, Xu & Cheng (2020)* |
| | Increase soil compaction | | |
| | Negative impact on soil fertility | | |
| | Leaching of heavy metals | | |
| Contaminants | Soil toxicity, terracummulation, leaching | Community structure and diversity | |
| | Negative impact on soil fertility | Increase of tolerance to contaminants and/or biodegradation | *Manzoni et al. (2012)*, *Tiemann & Billings (2011)* and *Leizeaga et al. (2020)* |
| | Enhance effects on antibiotics | Spread of antibiotic resistance | |
| | Repel soil water | | |

soil structure and water retention in general (*Wang et al., 2019*). This, however, is not a universal rule, and the reverse result can occur. In some cases, covering soil particles with organic matter or even microbial biofilms can increase the soil's hydrophobicity and thus its water repellence (*Nadav, Tarchitzky & Chen, 2013*). The fact that wastewater irrigation was sometimes associated with an increase in soil microbial biomass and soil enzyme activity suggests that organic matter supply has an impact on the microbiota (Table 4). Dehydrogenase, laccase, cellulase, protease, and urease activities are examples (*Chevremont et al., 2013*; *Morugán-Coronado et al., 2013*).

## Effects on microbial activity and abundance

Several studies have found that wastewater irrigation increases soil microbial biomass (Table 5). The simultaneous increase in the activity of dehydrogenase, a characteristic often indicative of biological oxidation of organic molecules (*Frenk, Hadar & Minz, 2014*; *Del Mar Alguacil et al., 2012*), suggests that this impact may be related to the availability of
**Table 5  Observed effects of wastewater (WW) irrigation on soil properties.**

| | | Effects on microbial parameters | | |
|---|---|---|---|---|
| **Soil description/Culture** | **Enzyme activity** | **Other activities** | **Implication for microbiota** | **Reference** |
| Vertisols/cereals and vegetables | DH | Denitrification activity | Increment of available P and water-soluble organic carbon is related with the increases of microbial biomass and activity | *Hoegh-Guldberg et al. (2018)* |
| | | Adenylate energy charge ratios | | |
| Xerorthent/orange-tree orchard | AP, BG, DH, PR,UR | Diversity of arbuscular mycorrhizal fungi | Reduction of the arbuscular mycorrhizal fungi diversity | *Hicks Pries et al. (2017)* |
| sandy-loam texture/mangrove swamp | AP, DH | Aerobic and anaerobic bacteria | Increment of available nutrients stimulates microbial growth and activity | *Hoegh-Guldberg et al. (2018)* |
| Leptosols/cereals and vegetables | DH | Denitrification activity | Increment of available nutrient easily decomposable organic increases soil microbial biomass and activity | *Li et al. (2019)* |
| Horticultural soil | Laccase, cellulose, PR, UR | Functional diversity of soil microorganisms | Increase of soil enzymatic activity is involved in the degradation of organic matter brought by water | *Allison, Wallenstein & Bradford (2010)* |
| Eutric Arenosol/lettuce | HD | Bacterial community structure | Ammonia-oxidizing bacterial community is stimulated by wastewater supply | *Frey et al. (2013)* |
| Loamy texture/Fodder, cereals | | Ammonia-oxidizing, Heavy metal resistant bacteria vesicular arbuscular mycorrhizae | Metal intrinsic endurance of bacterial and vesicular arbuscular mycorrhizae population is enhanced | *Allison, Wallenstein & Bradford (2010)* |
| Silty sand texture/perennial ryegrass | | Microbial abundance (total aerobic bacteria) | Low risk of microbial aquifer contamination | *Frey et al. (2013)* |
| Rhizosphere soil/wheat | | Metal resistant *Azotobacter chroococcum* isolates | Wastewater leads to an increase of metal resistance in rhizosphere *A. chroococcum* | *Jin, Xu & Cheng (2020)* |

organic carbon. Soil irrigation with wastewater is believed to stimulate diverse species and metabolic pathways by giving organic matter and nutrients. In soils irrigated with treated wastewater, increased activity of various enzymes (*e.g.*, hydrolytic, proteolytic, laccases, cellulases, phosphatase) has been observed (*Del Mar Alguacil et al., 2012*; *Adrover et al., 2012*). As a result, wastewater irrigation may boost the activity of microorganisms involved in the biochemical balance of elements including carbon, nitrogen, and phosphorus. Furthermore, organic matter, independent of soil microbial activity, may stabilize enzymes that stay active in the extracellular medium. However, these changes are not always beneficial, and increasing soil microbial abundance and activity may have detrimental consequences for soil characteristics. For example, *Li et al. (2019)* discovered that bacterial growth driven by wastewater irrigation resulted in the production of biofilms, which clogged the pore spaces between particles, affecting soil hydraulic conductivity. The complexities of the cause–effect interactions involved with wastewater irrigation are once again highlighted. In the available research, there is no broad agreement on how to increase microbial abundance and activity in wastewater-irrigated soils. Several publications have documented either a decrease in microbial biomass or no discernible effects on enzymatic activity (Table 5). In wastewater irrigated agricultural soil, (*Kayikcioglu, 2012*) found a decrease in the activity of the enzymes aryl sulfatase, dehydrogenase, urease, alkaline phosphatase, and -glucosidase. The fact that, in addition to nutrients, pollutants such as heavy metals, are delivered with wastewater irrigation is likely to explain the restriction of microbial growth or activity (*Kayikcioglu, 2012*). The complexity of the implications of wastewater irrigation makes it difficult to establish plausible cause–effect linkages, and providing organic matter is an excellent example. Although the majority of studies found an increase in organic matter as a result of wastewater irrigation, some found no significant differences but did find changes in microbiological and biochemical markers (Table 5).

## Valorization of Oil Mill Waste (OMW)

Every year, the olive oil business releases huge amounts of wastewater from olive mills (OMW). The chemical analysis of OMW reveals that they include a high concentration of phenolic chemicals, lipids, and organic acids, as well as a low biodegradability that limits their use (*Jeddi et al., 2016*). Different elements, the method of extraction, the technological process separation, the meteorological circumstances, and the variety and fruit age of the olive tree, all influence the properties of OMW (*Jeddi et al., 2016*). Because they are phytotoxic and can hinder plant growth, they should not be used for irrigation. Pretreatment of OMW can help solve these issues by improving the quality of the wastewater and removing some of its toxicity. To treat OMW, many treatments such as physical–chemical and biological processes are used. To treat OMW, dilution, evaporation, sedimentation, filtering, and centrifugation were mostly used as physical processes (*Khdair & Abu-Rumman, 2020*). Furthermore, decantation with lime and clay, coagulation–flocculation, electrocoagulation, natural evaporation, and thermal concentration have all been employed to lower the contaminating load of OMW (*Amor et al., 2019*). Evaporation ponds and artificial ponds with huge surface surfaces that are

designed to evaporate water efficiently using sunshine and ambient temperatures can also be utilized for OMW treatment (*Domingues et al., 2018*).

Plant development is stimulated by increasing levels of micronutrients, particularly potassium and organic matter (*Parađiković et al., 2019*). However, due to the high organic load of OMW, which is primarily attributable to the presence of polyphenols as well as short and long chain fatty acids, these effluents may have phytotoxic and antimicrobial effects (*Slimani Alaoui et al., 2016*). Microorganisms, particularly bacteria, may be inhibited from growing, which could slow down the mineralization process in the soil (*Slimani Alaoui et al., 2016*). As a result, the regulated spread of OMW can improve soil fertility and provide opportunities to recycle diverse chemicals.

According to *Dakhli et al. (2021)*, the addition of OMW sewage material to soil causes a decrease in pH and an increase in electric conductivity (EC). Water (83–96 percent), OM (3.5–15 percent), and mineral nutrients (0.5 to 2 percent) such as nitrogen, phosphorus, potassium, iron, and magnesium are abundant in OMW (*Malik et al., 2018*; *Geyer et al., 2019*; *Sinsabaugh et al., 2016*; *Utomo et al., 2013*; *Wieder, Bonan & Allison, 2013*). OMW alter microbial soil characteristics negatively, reducing or preventing microflora growth (*Hentati et al., 2016*). According to *Bombino et al. (2021)*, the OMW spreading harmed soil fertility by affecting physical and chemical soil properties. Other studies found that the long-term effect of OMW disposal on soil properties (*Bombino et al., 2021*) resulted in the effluent having a favorable influence on soil physical, chemical, and microbiological parameters. Some studies have shown that these wastes have severe impact on soil microbial communities, aquatic ecosystems, and even air quality (*Rajhi et al., 2018*). This understanding emphasizes the importance of assessing microbiological risk when disposing of olive mill residues. As a result, standards for managing these wastes through methods that minimize environmental impact and contribute to resource sustainability are required.

## CUE and stress in microbes

Numerous environmental factors influence microbial decomposition and CUE regulation, including the fungal-to-bacterial ratio, but microbial processes can also be influenced by stressors. Stress is defined as a change in the environment that poses physiological challenges to microbial function and survival (*Tiemann & Billings, 2011*), affecting CUE as a result of increased resource allocation to maintenance rather than growth (*Manzoni et al., 2018*). Previous research on the effects of stress on soil microbial processes has emphasized ecological stability theories. Numerous studies have used respiration and growth to assess soil functions because they reflect the effects of stress on physiological functions and how this affects major ecological processes such as decomposition (*Tiemann & Billings, 2011*).

Drought, freezing-thawing cycles, salinity, and environmental pollution are all common sources of stress in soils and water (*Rath & Rousk, 2015*). The presence of heavy metals in soil, as a result of industrial and mining activities, can cause acute or chronic stress to microorganisms (*Lucchini et al., 2014*). While heavy metals are necessary trace elements in biochemical reactions, they are unable to be synthesized or degraded by the cell, and thus persist in the environment. As a result, metal toxicity can occur at high concentrations,

 

the cell strictly regulates the influx, efflux, and intracellular concentrations of heavy metals (*Ladomersky & Petris, 2015*). Stress can cause changes in energy allocation and/or cell death in microorganisms during a brief stress event. By contrast, these changes probably occur to a lesser extent during long-term stress as a result of the community's selection of more tolerant groups. Different microbial groups have distinct strategies for coping with stress, and classifying them as fungi or bacteria makes numerous assumptions, but it can be a useful starting point for understanding resource allocation during stress events (*Yang et al., 2019*). Fungi have a lower surface-to-volume ratio than bacteria, which results in less contamination entering the cell, and fungal hyphae can also direct growth toward toxicant-free areas (*Le Gall et al., 2015*). As a result, it has been suggested that fungi are generally more resistant to metal stress than bacteria.

## CONCLUSIONS

Soil microbiomes provide essential ecosystem services such as conserving soil carbon and providing nutrients to plants, and their value in sustaining a healthy soil for future generations cannot be emphasized. The term CUE is a bit of a misnomer. Even the most basic definition, as the ratio of microbial biomass output to material intake from accessible substrates, is fraught with uncertainty because of disparities in estimation, temporal and geographical precision, as well as metabolic and taxonomic properties of the microbial population. CUE is unlikely to be consistent within or between systems because it is an emergent property of the system. It responds to changes in state and driving stimuli, but is bound by a number of biochemical and biophysical metabolic constraints. Shifts in microbial community composition and function will have ramifications for microbial interactions, biogeochemical cycling, which could exacerbate or mitigate climate change. The active SOC pool's residency period is determined by the soil microorganisms' C usage efficiency (CUE). CUE governs whether C is used to make enzymes for resource acquisition, make metabolites for chemical signaling and growth regulation, or create biomass. Temperature, substrate availability, and pH all have an impact on cell physiology and CUE. The CUE of species populations creates a collective response when they get together. Greater CUE occurs at the community level when biomass production is efficient (more biomass produced per unit of C digested), and some microbial communities have a stronger potential to store soil C than others.

As we learn more about the important roles played by microbes in soil ecosystems, we will be able to predict how environmental change will affect vital metabolic processes, as well as exploit this information to mitigate the detrimental effects of climate change. We recommend that CUE estimates within modeling frameworks take into account the study system's and aims' relevant aspects. Changes in resource composition, multi-resource stoichiometric restrictions, and microbial community physiology, as well as environmental causes, should be considered in models operating on finer time scales. For accurate comparison and application in future CUE research, it is critical to more clearly identify the specific research scale and process of microbial CUE.

## ACKNOWLEDGEMENTS

Special thanks to the College of Forestry, Gansu Agricultural University for providing a laboratory space for the conduct of experiments.

### Funding

This work was supported by "Research on the Coordination Relationship between Land Urbanization and Population Urbanization" (project No. GSAU-ZL-2015-046) and Fundamental Research Funds of Gansu Provincial Natural Science Fund of "Research on Land use and Ecological Security in Ecologically Vulnerable Areas" (project No. GSAN-ZL-2015-045). The funders had no role in study design, data collection and analysis, decision to publish, or preparation of the manuscript.

### Grant Disclosures

The following grant information was disclosed by the authors:
"Research on the Coordination Relationship between Land Urbanization and Population Urbanization" (project No. GSAU-ZL-2015-046).
Fundamental Research Funds of Gansu Provincial Natural Science Fund of "Research on Land use and Ecological Security in Ecologically Vulnerable Areas" (project No. GSAN-ZL-2015-045).

### Competing Interests

The authors declare there are no competing interests.

### Author Contributions

- Samuel Adingo conceived and designed the experiments, performed the experiments, analyzed the data, prepared figures and/or tables, and approved the final draft.
- Jie-Ru Yu conceived and designed the experiments, analyzed the data, prepared figures and/or tables, authored or reviewed drafts of the paper, and approved the final draft.
- Liu Xuelu conceived and designed the experiments, authored or reviewed drafts of the paper, and approved the final draft.
- Xiaodan Li performed the experiments, analyzed the data, authored or reviewed drafts of the paper, and approved the final draft.
- Sun Jing conceived and designed the experiments, analyzed the data, prepared figures and/or tables, and approved the final draft.
- Zhang Xiaong analyzed the data, prepared figures and/or tables, and approved the final draft.

### Data Availability

Table 1: Different microbial carbon use efficiency measurement methods
Table 2: Variation of microbial carbon use efficiency
Figure 1: Microbial metabolic components and equilibrium equation.

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
