# Peer review of "Variation of soil microbial carbon use efficiency (CUE) and its Influence mechanism in the context of global environmental change: a review"

_PeerJ, doi:10.7717/peerj.12131_

## Round 0.1 · original submission · Major Revisions

Reviewers have now commented on your paper. You will see that they have requested substantial changes (major revision) to be made. If you are prepared to undertake the work required, I would be pleased to reconsider my decision.

Reviewer 1 ·

Basic reporting

The review is dealing with soil Microbial Carbon Use Efficiency (CUE) variation as a result of Global Environmental Change.
CUE is an important concept for understanding the future trajectory of soil-climate feedbacks which was not sufficiently reviewed. CUE is thus increasingly recognized by microbiologists, ecologists, and modelers alike as essential for understanding the causes and consequences of microbial C cycling. Accordingly, the present review is interesting to understand the consequences of environmental change on CUE and C cycling in general.
Despite the importance of the topic covered in this review, however, there are many drawbacks in this work.
First, the review was not well organized and could be improved in term of structure and content. In fact, CUE reflects a collection of numerous processes (physiological, ecological and community dynamics) that influence C metabolism across varying scales of time and space. The responses of these processes to environmental changes are very important to understand the relevance of CUE at different scales from a single microbial cell to entire ecosystems. However, these processes were not well presented in this review. For example, the environmental sensitivity of microbial physiology and microbial community structure and activity are very important to determine the impact of environmental changes on CUE. However, authors have not well developed these aspects in their review and instead they focused on quantifying CUE methods which were previously reported in many reviews.
Figures and tables should be added to better illustrate the impact of environmental changes on CUE and to give more information about the possible interaction between the most influencing factors and microbial soil communities.

Experimental design

CUE depends on a complex processes linked together at different scales which makes the study of the CUE variation a challenging task. Thus, the impact of each environmental changes on these processes should be investigated separately. Authors are invited to firstly define CUE and indicate the parameters influencing microbial carbon metabolism in relation to 1) soil characteristics (pH, soil moisture, nutrients availability, organic matter composition, organo-mineral interaction, physical properties such as porosity ...) and 2) soil microbial community structure and dynamic to determine the interaction between different microbial groups involved in C cycling. Soil microbial activity change from one ecosystem to another according to carbon source, microbial community and ecological environmental factors. For instance, CUE is not the same in humid and semi arid zones. Thus, authors should highlighted the variation of CUE in different ecosystems and climatic zones as many researches were conducted on these aspects. Investigating the impact of variation of global environmental change is more difficult as it is a dynamic process requiring mathematical models to analyze and to predict the evolution of CUE which depends on the interaction between different biological and physicochemical parameters at two different scales (time and space).

Validity of the findings

According to the purpose indicated in the introduction, Limits of ecological environmental factors and Composition of microbial community are the most important sections of the review. Authors are invited to more develop these two sections and to analyze the possible correlation between the environmental factors parameters on CUE and C cycling. Furthermore, a section on soil microbial adaptation and the impact of environmental factors on the dynamic and diversity of soil microbial communities should be added.
The conclusions were not well linked to the purpose of this review.

Reviewer 2 ·

Basic reporting

Reviewers' comments:

This manuscript entitled “Variation of soil microbial carbon use efficiency (CUE) and its Influence mechanism in the context of global environmental change” has shown general description of CUE and its measurement methods, also reviewed and analyzed the research progress of soil microbial CUE variation and influencing factors. In addition, it provides a theoretical foundation for scientist, researchers and relevant stake- holders to predict future climate change. Therefore, it needs to be further modified to improve the quality of this article.
The main concerns about the paper are as follows:

Experimental design

1. The sentence: “Water stress, for example, inhibits the …..”. Author should mention some detailed researchers on the spreading of wastewater on the soil.
2. In the other article paragraphs the author does not give details of the articles cited in effect of water in CUE. We would have liked to see some impacts of using wastewaters by providing a section that discussion the effect on microbial composition, CUE, substrate intake. Also, include in the review work; recently published work related to valorization of OMW
3. For the subject some references are old and not all them are very relevant, use newer and at least highly cited ones.
4. We would have liked to see representative table citing different carbon sources (acids, sugars, lignin, phenol, etc.) on the CUE.

Validity of the findings

None.

Additional comments

1. Line 70: interactions interactions (delete repeated word).
2. Line 74: Full stop for the sentence “sequestration in soil ecosystems.”
3. Line 141 “and methods based…” to Line 142: too long sentence.
4. Author should add citation in the section of Substrate absorption rate method and Substrate concentration change method.
5. Line 261 Delete repetitive references
6.Line 356: soil organic carbon (12.3). (18.2-75)….. More explanation??

---

## Round 0.2 · accepted · Accept

Your paper is accepted without any need to add the suggested citations.

Reviewer 2 ·

Basic reporting

1. Authors should enhance the figure resolution.
2. Line 647: surface surfaces (delete repeated word).

Experimental design

1. Author should include in the review work; recently published work related to valorization of OMW: a. https://doi.org/10.1002/cbdv.201900608 ; b. https://doi.org/10.1007/s13762-021-03145-0
2. Divide your article into clearly defined and numbered sections like numbered 1.1 (then 1.1.1, 1.1.2, ...), 1.2, etc so that the reader can follow and understand the tenor of the paper.

Validity of the findings

Good.